# Modeling of the Closure of Metallurgical Defects in the Magnesium Alloy Die Forging Process

**DOI:** 10.3390/ma15217465

**Published:** 2022-10-25

**Authors:** Grzegorz Banaszek, Teresa Bajor, Anna Kawałek, Marcin Knapiński

**Affiliations:** Faculty of Production Engineering and Materials Technology, Czestochowa University of Technology, 42-201 Czestochowa, Poland

**Keywords:** magnesium alloy AZ91, physical modeling, forging, closure of discontinuities

## Abstract

The article discusses the impact of hot forging elongation operations on the closure of metallurgical discontinuities such as middle porosity in selected magnesium alloys (AZ91) depending on the shape of the input used. Numerical modeling was carried out using the Forge^®^NxT 2.1 program based on the finite element method and laboratory modeling in order to bring about the closure of defects of metallurgical origin in deformed forging ingots. On the basis of the conducted research, optimal values of the main technological parameters of forging and appropriate groups of anvils to be used in individual stages of forging were proposed in order to eliminate metallurgical defects.

## 1. Introduction

The search for lightweight construction materials characterized by favorable strength parameters is still a leading topic among the scientific community. In the current political and economic situation, with turbulence in the energy market, reducing the weight of cars and fuel consumption and reducing the impact of greenhouse gases emitted by cars is an important element for consumers [1,2,3]. Magnesium alloys, as the lightest construction materials and showing good heat dissipation and vibration damping, are gaining and finding more and more applications in the automotive industry [4,5,6,7,8,9,10,11,12]. A large number of magnesium alloy products are obtained mainly in extrusion and stamping processes, less often in rolling and forging processes. We should pay attention to forged products made of magnesium alloys due to their homogeneous microstructure and improved mechanical properties compared with cast alloy elements. Designing forging technology requires a comprehensive approach to the research problem posed. High variability of shaping parameters such as: temperature; sequence of operations and technological treatments, for example, turning the material and applying deformations using anvils during forging elongation operations; values of applied reductions; relative feed values; deformation speed; and shape-dimensional parameters of the working surfaces of anvils makes obtaining forgings made of high-quality magnesium alloys with a homogeneous microstructure throughout the whole volume an extremely difficult matter. Due to the hexagonally compact crystallographic structure, magnesium alloys have limited plasticity and poor deformability at ambient temperature. The correct implementation of the free-forging process in flat and shaped anvils of selected magnesium alloys allows obtainment of a product with satisfactory final properties [13,14]. Shaping products by forging methods provides a method to close metallurgical discontinuities formed at the stage of production of the input material. In the case of forming high-quality products, especially in terms of the absence of metallurgical discontinuities, the application of the lengthening operation is, according to the authors, an innovative method of charge preparation, which enables their liquidation. The elongating operation in flat anvils creates such a state of stress that favors the welding of metallurgical discontinuities. The analysis of methods of closing metallurgical defects in the cross-section of forgings is presented in the works [15,16,17,18,19,20,21], where it was shown that the closure of metallurgical defects in deformed forgings is influenced by the main parameters of the forging process, such as reduction, relative feed, charge temperature, and the shape and dimensions of anvils.

## 2. Test Objective and Scope

The aim of the study was to investigate the influence of elongation operations on the closure of metallurgical discontinuities of the middle porosity type depending on the shape of the applied input.

In order to realize the aim of the work set out in this way, the authors proposed that hot forging elongation operations should be carried out using flat anvil assemblies.

In order to achieve the goal of the proposed work, numerical studies of elongation operations in flat anvil assemblies for two geometric shapes of samples were carried out.

Modeling of elongation operations was performed using Forge^®^NxT 2.1 based on FEM, which allows tracking changes in temperature distributions, hydrostatic pressure, and strain intensity in the plastically processed magnesium alloy AZ91. The distributions of temperature values, hydrostatic pressure, and strain intensity over the cross-section of the deformed alloy after each forging pass were determined.

During tests of forging elongation operations, the results of plastometric tests on a given AZ91 alloy were used, on the basis of which stress–strain dependence diagrams of the actual alloy were developed and the coefficients of the plasticizing stress function were selected [13,14]. The ranges of deformation, deformation speed, and temperature changes during the theoretical research were assumed on the basis of the characteristics of forging machines used in real forging processes, and on the basis of literature data and the authors’ own research carried out at the Department of Plastic Processing and Safety Engineering of the Czestochowa University of Technology [13,14,15,16,17,18,19,20,21,22,23,24,25,26].

In order to compare the results of numerical modeling with the actual laboratory process, the samples were deformed using the Gleeble 3800^®^ metallurgical processes simulator. On the basis of the conducted research, an analysis of the influence of the geometric shape of the input on the welding of the modeled internal metallurgical discontinuity of the middle porosity type was carried out. Physical verification of the numerical model was also carried out to determine the structure of the forged magnesium alloy AZ91 in the discontinuity modeled zone and a comparison of this structure with the discontinuity zone.

The results of the tests should contribute to improving the structural and mechanical properties of AZ91 alloy bars through the appropriate selection of the shape of the feedstock.

## 3. Materials and Methods

The material chosen for the tests was magnesium alloy AZ91 with a chemical compositions as provided in Table 1.

## 4. Methodology of Numerical Research

The elongation operation in flat anvils of four samples made of magnesium alloy AZ91 was analyzed. Two samples were in the shape of a cube measuring 10 × 10 × 10 mm, and two more were in the shape of a cylinder with a diameter of 10 mm and a length of 10 mm. In two samples of different shapes, an axial internal discontinuity was modeled by making a hole in the axes of the samples with a diameter of 2 mm equal to their length. For the purposes of numerical modeling, the sample model was made in the AutoCad 2009^®^ computer program and the designed discontinuity was treated as the difference in volume between the sample and the discontinuity.

To model the elongation operation, a commercial computer program on PC FORGE^®^NxT 2.1 was used, which is a product of Transvalor Solution, based on the finite element method (FEM) [27]. This program allows for thermomechanical simulation of, among others, plastic processing. A detailed description of temperature, energy, strain, and deformation functions as well as thermomechanical and friction laws, used during the calculations, can be found in the papers [13,14,16]. Calculation of thermal properties was made by using the Galerkin equation, while the strengthening curves were approximated by the Hensel–Spittel equation [28]. In the paper, to simulate the elongation operation, a thermo-viscoplastic model of the deformed body, which is based on the theory of large plastic deformations, was used. To generate the grid of finite elements, tetrahedral elements with the base of triangles were used. In the generated model input, the number of nodes equal to 3369 was used for simulation, while the number of tetrahedral elements adopted for the simulation was 30,084. In addition, in the axial area where there was a modeled discontinuity due to the mesh compaction tool in the program, 9116 nodes were generated, which accounted for 58,726 elements. This was done in order to increase the accuracy of calculations and to more accurately illustrate the mechanism of discontinuity welding, which then affected the accuracy of the obtained results and the subsequent analysis of these results. The value of the coefficient of friction between the surface of the anvils and the deformed rod was determined to be μ = 0.3 according to Coulomb’s law. It was assumed that the heat transfer coefficient between the anvils and the material is h = 10,000 W/m^2^K, while the heat transfer coefficient between the metal and the environment is equal to h = 10 W/m^2^K. The ambient temperature was assumed to be equal to 20 °C, while the temperature of the anvils was equal to 300 °C. The initial temperature of the input before the elongation operations in its entire volume was assumed to be the same and equal to 400 °C. The feed speed of the upper anvil was equal to v = 8 mm/s, while the lower anvil was assumed to be stationary in all forging passages for all four samples. During numerical modeling, all four samples were deformed with a relative reduction of 35%, then rotated clockwise by an angle of 90° and deformed again with a reduction in the same value. The same boundary and initial conditions were assumed during physical modeling carried out in the Gleeble 3800^®^ metallurgical processes simulator.

## 5. Analysis of Distributions of Temperature Values, Effective Strain, and Hydrostatic Pressure during the Elongation Operation of a Circular Sample

The results of the tests concerning the distribution of temperature values, effective strain, and hydrostatic pressure values during the elongation operation of samples with a circular cross-section made of magnesium alloy AZ91 are shown in Figure 1, Figure 2, Figure 3, Figure 4, Figure 5, Figure 6, Figure 7, Figure 8, Figure 9, Figure 10 and Figure 11.

The data in Figure 1 show that in the axial zone of the sample, the temperature after deformation was 386 °C, it is a decrease of 14 °C compared with the initial forging temperature, which is 400 °C. In the areas to the right and left of the sample, the temperature was much higher at 388 °C. The material flowed freely in these zones, unrestricted by the working surfaces of the anvils, and despite contact with the environment at the assumed temperature of 20 °C, it did not cool significantly due to a large deformation work, which influenced maintaining the temperature close to the initial temperature. In the upper and lower part of the sample, along the *y*-axis, there was a temperature drop of 10 °C from the initial temperature. This was the result of heat transfer to the anvils, which were heated before being deformed to 300 °C. The anvils were heated to such a temperature that the heat flow from the sample toward the working surfaces of the anvils was not significantly large. In the actual conditions of industrial forging, the anvils are heated to a temperature range of 200–300 °C before the elongation operation, so that the forged material does not cool down too quickly in subsequent forging transitions. The first anvil transition is not heated because it maintains its temperature due to the flow of heat from the deformed material in subsequent transitions.

By analyzing the data presented in Figure 2, it can be stated that after rotating the sample by 90° and deforming again with a relative reduction of 35%, in the middle part of the sample along the *x* axis there was a temperature drop of 26 °C from the initial temperature. Despite the large work of deformation caused by the significant reduction, the temperature drop was due to the longer duration of the second forging transition. The longer duration of the second forging transition was associated with the rotation of the sample by an angle of 90°. In the zones of the material lying under the working surfaces of the anvils, a temperature equal to 367 °C was recorded, this is a decrease of 33 °C from the temperature at the beginning of forging process.

Figure 3 and Figure 4 compare the distributions of the effective strain after deformation of the sample with the modeled discontinuity (Figure 3) and without the modeled discontinuity (Figure 4). The data presented in these figures show that in the axial zone of the sample with modeled discontinuity, there was a disturbance in the deformation intensity distribution. This was due to the intensification of deformations within the welded discontinuity. The shape of the welded discontinuity (the hole inside the cylinder-shaped sample) influenced the different nature of the metal flow in the zone of its occurrence; it was related to different directions of the friction and pressure force vectors, which were initiated by a change in the geometric shape of the welded discontinuity during the growing reduction that occurs during deformation. In the sample without discontinuities (Figure 4) this phenomenon was not observed, only the external shape of the deformed sample and the shape of flat anvils, which forced such and no other flow of the deformed alloy, were responsible for the directions of the vectors of friction forces and pressure. The deformation intensity values in the axial zone of the sample with discontinuity (Figure 3) ranged from 1.20 to 1.50, and in the same zone in the sample without discontinuities (Figure 4) were within the range of 0.70–0.90. In other zones outside the area of occurrence of the welded discontinuity in both cases, the distribution of the deformation value was of the same nature and its value was in the range of 0.19–0.62.

By analyzing the data in Figure 5 and Figure 6, it can be stated that the distributions of the hydrostatic pressure values do not show such intense changes in the metal flow in the discontinuity zone compared with the sample without discontinuities (Figure 3 and Figure 4). There is a difference in the axial zones of both samples, but the difference is not significant and amounts to 17 MPa. In other sample zones, the distribution of hydrostatic pressure values is the same, which indicates that the welded discontinuity in the axial zone does not affect the metal flow in the other zones of the deformed sample.

Based on the data shown in Figure 7, it can be stated that in the sample of the magnesium alloy AZ91 within the welded discontinuity, there is a different nature of the effective strain distribution. In the areas located on the right and left side of the discontinuity, there are large values of effective strain of the order of 1.50, and in the further distance, the values are in the range of 0.91–1.20. The data presented in Figure 7 show that the effective strain values reach the maximum values and these are compressive deformations. In the areas located in the upper and lower part of the welded discontinuity, the effective strain values are small and fluctuate in the range of 0.33–0.62, these are tensile deformations. In order to completely weld discontinuities, it is important to select the main technological parameters of forging operations and a shape of dies or anvils to ensure the greatest possible intensification of compressive deformations within the welded discontinuities during the deformation of forgings. Within the welded discontinuity, such a distribution of effective strain is undesirable where tensile deformations prevail. Such a deformation distribution blocks the process of welding discontinuities in forging processes and therefore the final product will settle defects in the form of unwelded discontinuities of the middle porosity type.

Figure 8 shows the distribution of hydrostatic pressure around the welded discontinuity during sample deformation at a reduction of 20%. The presented data show that in the areas on the right and left side of the welded defect, there is a high hydrostatic pressure with values from 83 to 117 MPa—these are compressive stresses. However, there is no hydrostatic pressure in the areas above and below the welded discontinuity. There are tensile stresses of 50 MPa, which is 50% lower than the values of compressive stresses (right and left side, 117 MPa). This distribution of hydrostatic pressure values is advantageous in terms of discontinuity welding. Equalization of the values of tensile and compressive stresses or higher values of tensile stresses in the welded discontinuity zone block their welding, therefore discontinuities in the final product in the form of middle porosity remain.

Figure 9 shows an axonometric projection of a deformed sample with 35% reduction with a completely unheated discontinuity visible inside. A view of this discontinuity is shown due to the fact that in the figures showing the surface of the transverse cross-sections, this discontinuity is poorly visible. The presented discontinuity is not welded because the applied reduction was insufficient. The use of larger reductions was impossible because it is known from forging practice that the use of one-time reductions above 35% causes cracks in materials along the axis of deformed forgings.

By analyzing the data in Figure 10, it can be stated that in the axial zone of the deformed sample and in areas slightly distant from the axial zone, the effective strain values were large and amounted to 1.50. During the second reduction, after prior rotation of the sample by 90°, the axial discontinuity was completely welded due to the high intensity of deformations occurring in this zone. In the area outside the axial zone of the deformed sample, different values of effective strain were recorded, ranging from 0.04 to 0.77.

The analysis of the data in Figure 11 shows that in the axial zone of the deformed sample, after the second reduction, the hydrostatic pressure values were in the range of 50–67 MPa and it was high enough to lead to complete welding of the discontinuity occurring there. In the area outside the axial zone, the hydrostatic pressure values were very small, and in some small areas, there was no hydrostatic pressure.

## 6. Distributions of Temperature, Effective Strain, and Hydrostatic Pressure Values on the Surface of the Cross-Section of the Deformed Sample with a Square Cross-Section Made of AZ 91 Magnesium Alloy

The test results on the distribution of temperature values, effective strain, and hydrostatic pressure values during the elongation operation of the AZ91 magnesium alloy square sample with modeled axial discontinuity are presented in Figure 12, Figure 13, Figure 14, Figure 15, Figure 16 and Figure 17. The data obtained during the elongation of a sample with the same cross-section without the modelled discontinuity were not presented because the obtained data were identical to the data for a sample with discontinuity. This was due to the fact that for a sample with a square cross-section, a total discontinuity welding was obtained after the first final bend of 35%, so there were no differences in the nature of the obtained distributions.

The data in Figure 12 shows that in the middle area lying along the *x*-axis of the deformed sample, the temperature value was 378 °C and was lower by 22 °C than the initial value. In the tool–material contact zones, the temperature value was 371 °C. It is worth noting that after the first forging transition in a sample with a circular cross-section (Figure 1), the temperature drops were lower and amounted to about 15 °C and thus, in the axial zone, the drop was 14°, and at the point of contact the material–tool contact zone drop was 10°. The comparison of data provided in Figure 12 with the data from Figure 1 shows that the shape of the hot forging input material, especially the input material with small initial dimensions, has a large impact on the temperature distribution in the volume of forgings. During deformation, the cylinder-shaped forging cools down more slowly than the cube or cuboid-shaped forging. This is due to the different nature of heat transferring to the environment, and in particular, heat transferring to the anvils. The deformed cylinder-shaped sample in the initial phase of pressure on the flat anvils only has a point contact with the upper and lower anvils. In later stages of the operation, this contact gradually increases until it reaches the final reduction of 35%. After applying a 20% reduction, the entire upper and lower surface of the sample will be in contact with the anvils. As a result, there will be a smaller temperature drop in individual zones of the deformed sample when torn with a cube-shaped sample. For this sample, in the first phase of the operation, the upper and lower surfaces of the sample are in complete contact with the working surfaces of the anvils, hence a much greater temperature drop in the volume of the deformed sample occurs.

By analyzing the data in Figure 13, it can be stated that in the central area of the sample lying along the *x*-axis, after the second reduction, the temperature was equal to 365 °C, which is a decrease of 35 °C from the initial temperature. In the tool–material contact zone, a temperature equal to 358 °C was recorded, which means that in these zones the temperature fell by 42 °C from the initial temperature.

Figure 14 shows the distribution of the effective strain values on the cross-sectional area of the sample after the first reduction. The data presented in this figure show that in the central area of the sample, the values of effective strain were obtained within the range of 0.62–1.06, while in the external areas, the values were obtained within the range of 0.04–0.77. The geometric shape and initial dimensions of the sample, deformed in the elongation operation in flat anvils, have not only a strict impact on the distribution of temperature values, but also on the distribution of the effective strain values, causing different directions of the vectors of friction forces and pressure, which are responsible for the kinematics of the flow of the magnesium alloy in question. In this case, the shape of the sample (cube) had a beneficial effect on the welding of the axial internal discontinuity because a magnesium alloy flow kinematics was obtained, which, after deformation with a 35% reduction, allowed the discontinuity to be welded in the first forging transition. This was not achieved in the first forging transition with the same technological parameters and the same anvil of the sample in the shape of a cylinder.

The data in Figure 15 show that in the central area of the sample, large values of effective strain were obtained, which ranged from 1.35 to 1.50. If the metallurgical discontinuity was not welded in the first forging transition, the large value of the obtained deformations in the second transition would lead to the discontinuity being welded. In the outer areas of the sample, after the second transition, the effective strain values were very diverse in practically the entire range of values given in the legend. For the analyzed cross-sectional area of the sample, a forging cross is visible, characteristic for conducting a number of elongation operations in flat anvils, in particular for batch materials with a square or rectangle cross-section. This introduces large inequalities in the distribution of values, not only of deformations but also stresses, and not only in forged magnesium alloys, but also in many other alloys found in iron alloys. This is inevitable during forging of flat anvils, but the advantage of the described forging cross is the intensification of deformations and stresses in the axial forging zones where there is an axial discontinuity of the middle porosity type that is difficult to level in forging processes.

By analyzing the data in Figure 16, it can be stated that in the central area of the sample there were high hydrostatic pressure values in the range of 67–117 MPa, while in the outer areas on the right and left side of the deformed sample, after the first forging transition, the hydrostatic pressure values were small and ranged from 0.4–17 MPa. In the axial zone of the sample, large hydrostatic pressure values occurred, which allowed the axial discontinuity to be welded. In this case, it was possible to ensure this due to the task of maximum applicable relative reduction with the use of flat anvils, which introduce stress intensification directed to the axis of the deformed material.

The data in Figure 17 shows that after rotating the sample by 90° and re-executing the relative reduction of 35%, in the internal area of the sample, small hydrostatic pressure values were recorded, which were in the range of 17–33 MPa. While in the areas located on the left and right side of the sample along the *x*-axis, the pressure values ranged from 0.4–12 MPa. After the sample was rotated and reduced once again with the same value, large hydrostatic pressure values were not obtained in the axial zone of the sample. Hence, it is important to determine the appropriate deformation parameters which make it possible to achieve complete welding of axial discontinuities in the initial stages of forging, where the material is still very plastic and the temperature in the sample axis is still close to the initial forging temperature.

## 7. Physical Modeling of Closing Discontinuities in Forging Conditions in Flat Anvils

In order to verify the numerical tests of the material elongation process with artificially introduced axial discontinuity, physical simulations were carried out using the MaxStrain module of the Gleeble system. Two types of samples were used for the tests: with a square cross-section of 10 × 10 mm and with a circular cross-section with a diameter of 12 mm. In both samples, holes were made along the axis of the samples with a diameter of 2 mm. The holes were a model image of metallurgical axial discontinuities appearing in the cast input material intended for elongation operations. Figure 18 shows samples prepared for deformation.

The samples were heated to 400 °C, maintained at this temperature for 150 s, and then deformed in flat anvils with a 35% reduction, reducing the initial height from 10 mm to 6.5 mm (for a square section sample) and from 12 mm to 7.8 mm (for a circular section sample). After the set deformation, the samples were heated to a temperature of 400 °C, rotated by an angle of 90°, and then another deformation was set in such a way as to obtain a sample with a final height of 10 mm (for a sample with a square cross-section) and a height of 12 mm (for a sample with a circular cross-section) from the material expanded after the first deformation. In both cases, a constant anvil speed of 8 mm/s was used.

After deformation, the samples were cut in a plane perpendicular to their axis in the middle of their length. On the obtained cross-sections, metallographic micro-sections were prepared using the HNO_3_ nitric acid reagent (65%) and C_2_H_5_OH ethanol (96%) digestion. The micro-sections were subjected to observation using optical microscopy and scanning electron microscopy. The aim of the observation was to analyze the structural changes in the material occurring during deformation in the area of the modeled discontinuity. Figure 19 and Figure 20 show images from the optical microscope of the disclosed sample microstructures after deformation. Figure 19a shows a sample with a square cross-section in which no artificial axial discontinuity was introduced, while Figure 19b shows a deformed sample in which there was initially an artificially introduced discontinuity. However, Figure 20a,b show analogous areas disclosed in a sample of a circular cross-section.

Since the observation with the use of an optical microscope did not show discontinuities in the deformed samples, which indicates that during the plastic deformation the introduced axial discontinuity was closed, the microstructure of the material in the closed zone of the defect was additionally observed using a scanning electron microscope. Figure 21 and Figure 22 show microstructure images obtained from SEM analysis for a closed discontinuity area at two different magnifications. Figure 21 shows the microstructure of a sample with a square cross-section, while Figure 22 shows a sample with a circular cross-section.

The conducted SEM analysis confirmed the previous observations carried out with the use of an optical microscope, which concluded that the deformation system used is sufficient to completely close the axial metallurgical discontinuity in the AZ91 alloy.

## 8. Conclusions

Based on the analysis of the results of the conducted research, the following final conclusions were drawn:

The geometric shape of the input material significantly affects the welding of the internal metallurgical discontinuity of the type of central porosity.

As a result of the conducted tests, it was found that the complete welding of the axial discontinuity was obtained in a cylinder-shaped sample after the first forging transition, while it was obtained in a cube-shaped sample after the second forging transition.

Large hydrostatic pressure values achieved during the elongation of the AZ91 magnesium alloy samples in the first two forging transitions positively affected the welding of metallurgical discontinuities.

The use of a single, possibly large relative reduction influences the formation of a deformed sample of stresses and compressive deformations in local areas, which are conducive to the sealing of axial discontinuity.

During the conducted physical research, the authors observed that in the range of temperatures of the deformation process, there was no cracking of the material in its local zones and no adhesion of the outer layers of the magnesium alloy to the working surfaces of flat anvils.

## Figures and Tables

**Figure 1 materials-15-07465-f001:**
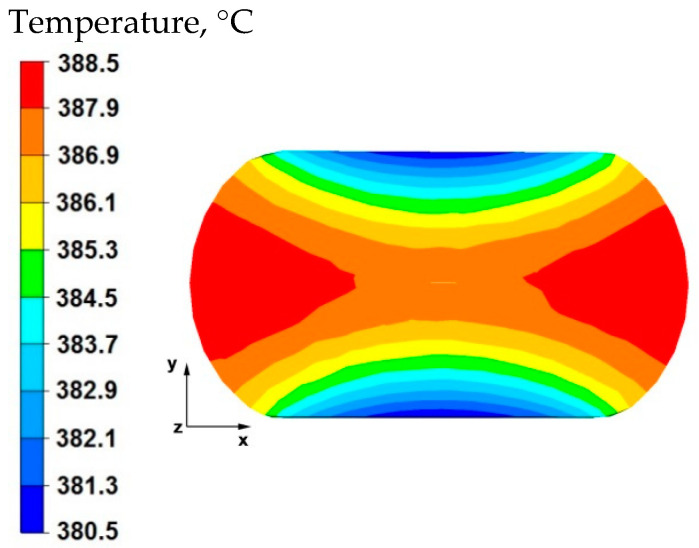
Distribution of temperature values on the surface of the cross-section of the sample with modeled discontinuity—after deformation with 35% reduction.

**Figure 2 materials-15-07465-f002:**
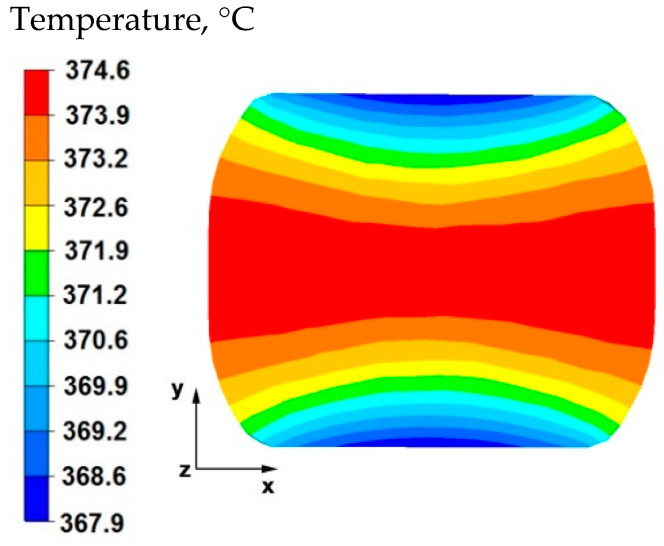
Distribution of temperature values on the surface of the cross-section of the sample with modeled discontinuity after rotation by 90° and re-deformation with 35% reduction.

**Figure 3 materials-15-07465-f003:**
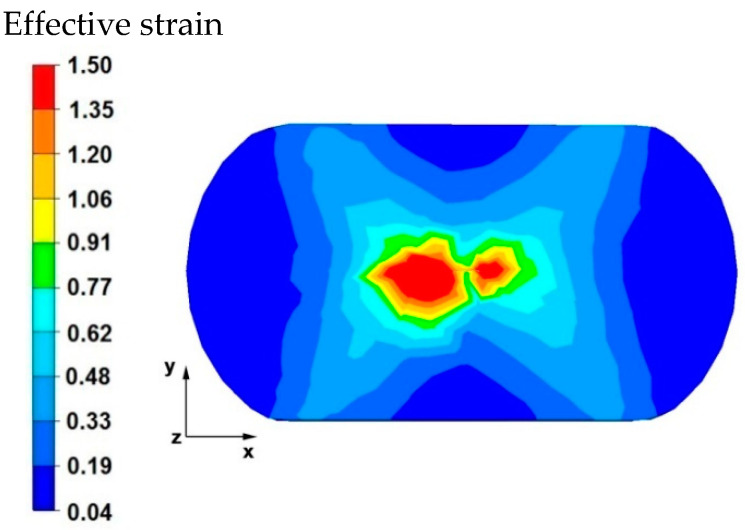
Distribution of the effective strain values on the surface of the cross-section of the sample with modeled discontinuity after deformation with 35% reduction.

**Figure 4 materials-15-07465-f004:**
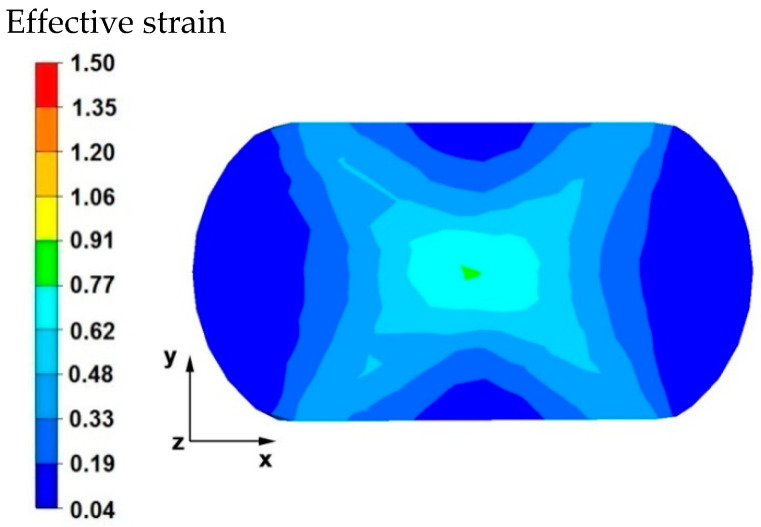
Distribution of the effective strain value on the surface of the cross-section of the sample without the modeled discontinuity after deformation with 35% reduction.

**Figure 5 materials-15-07465-f005:**
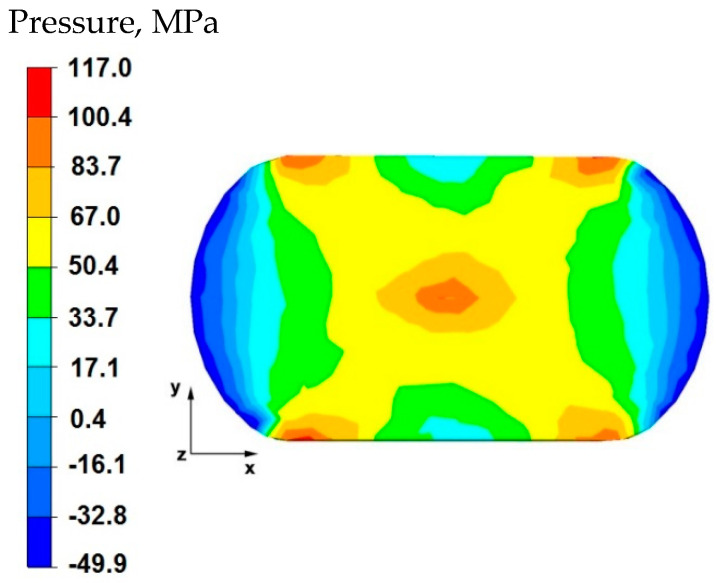
Distribution of hydrostatic pressure values on the surface of the cross-section of the sample with modeled discontinuity—after deformation with 35% reduction.

**Figure 6 materials-15-07465-f006:**
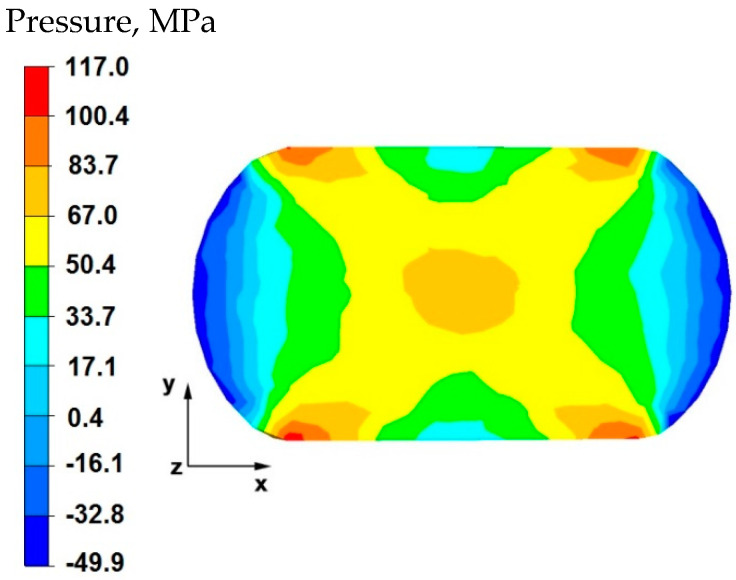
Distribution of hydrostatic pressure values on the surface of the cross-section of the sample without modelled discontinuity—after deformation with 35% reduction.

**Figure 7 materials-15-07465-f007:**
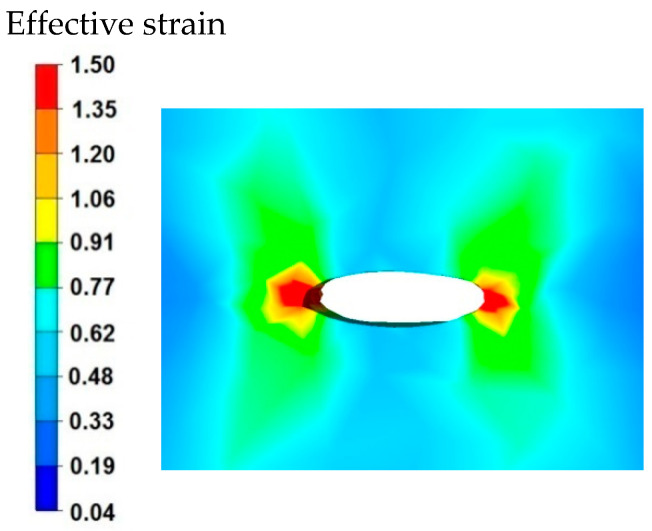
Distribution of effective strain values within the modelled discontinuity at a 20% reduction.

**Figure 8 materials-15-07465-f008:**
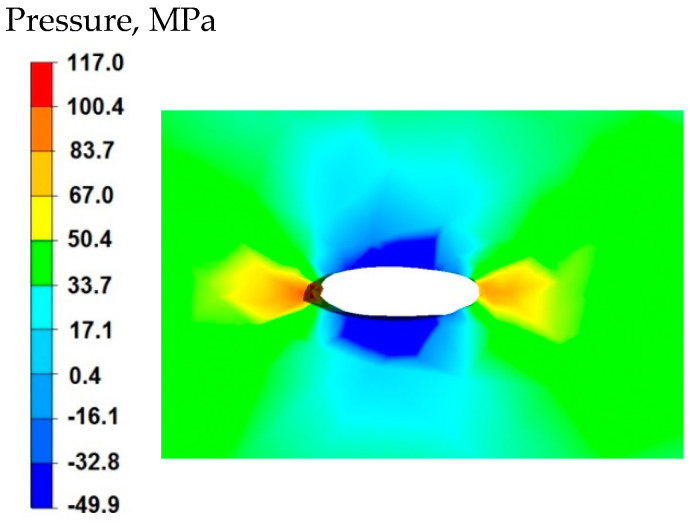
Distribution of hydrostatic pressure values within the modeled discontinuity at a 20% reduction.

**Figure 9 materials-15-07465-f009:**
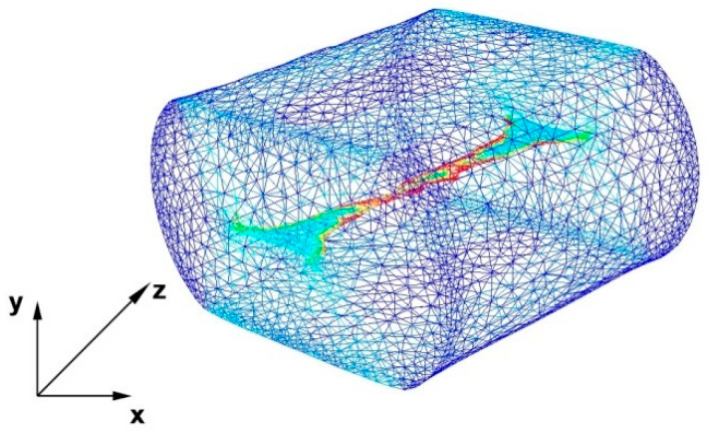
View of an unheated fully modeled discontinuity intended to simulate the central porosity inside a circular sample deformed with 35% reduction.

**Figure 10 materials-15-07465-f010:**
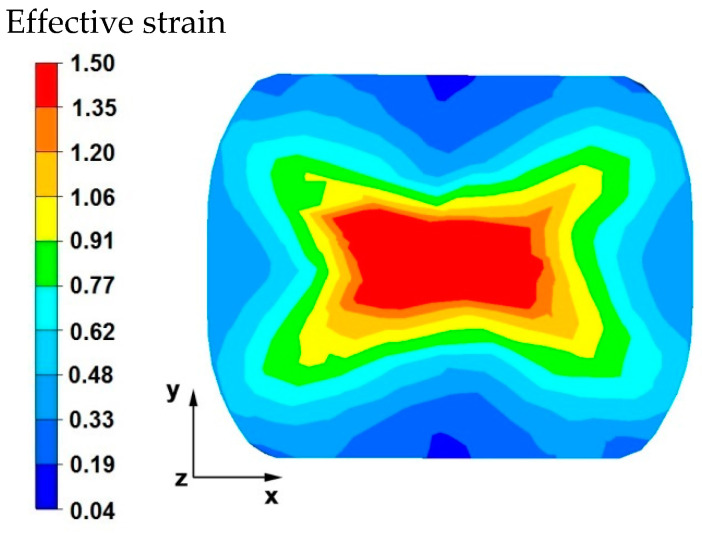
Distribution of the effective strain values on the surface of the cross-section of the sample with modeled discontinuity after rotation by 90° and re-deformation with 35% reduction.

**Figure 11 materials-15-07465-f011:**
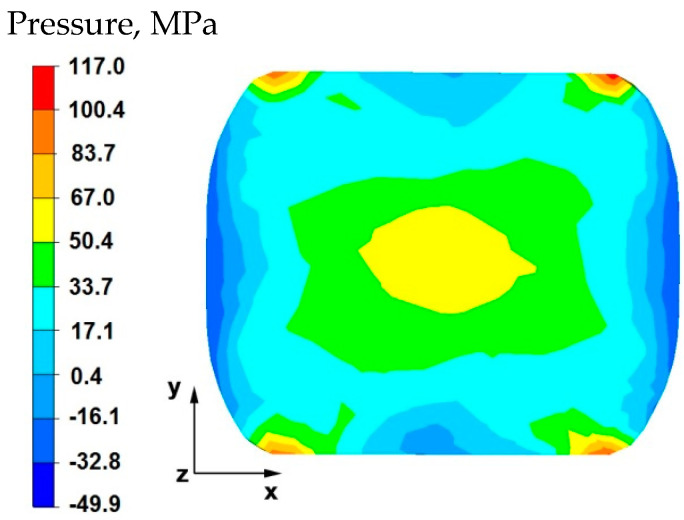
Distribution of hydrostatic pressure on the surface of the cross-section of the sample with modeled discontinuity after rotation by 90° and re-deformation with 35% reduction.

**Figure 12 materials-15-07465-f012:**
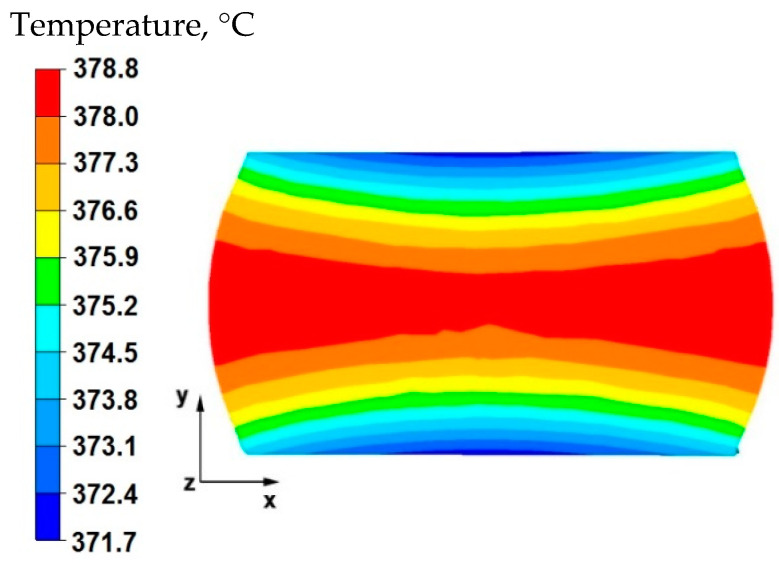
Distribution of temperature values on the surface of the cross-section of the sample with modeled discontinuity after deformation with 35% reduction.

**Figure 13 materials-15-07465-f013:**
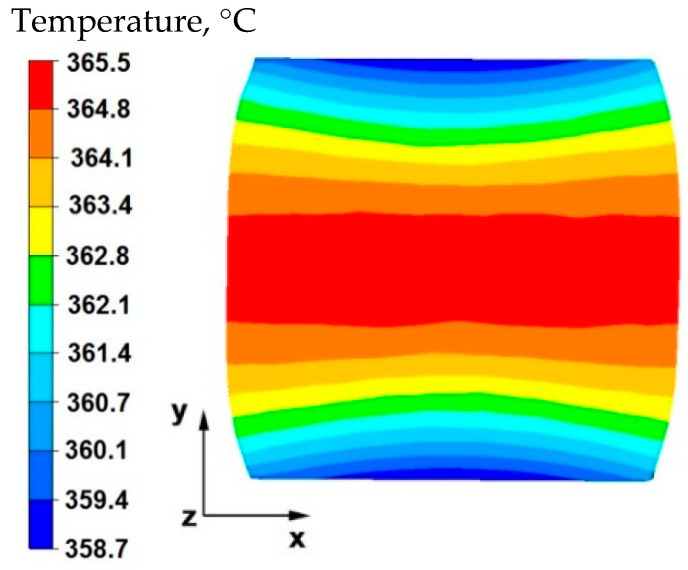
Distribution of temperature values on the surface of the cross-section of the sample with modeled discontinuity after rotation by 90° and re-deformation with 35% reduction.

**Figure 14 materials-15-07465-f014:**
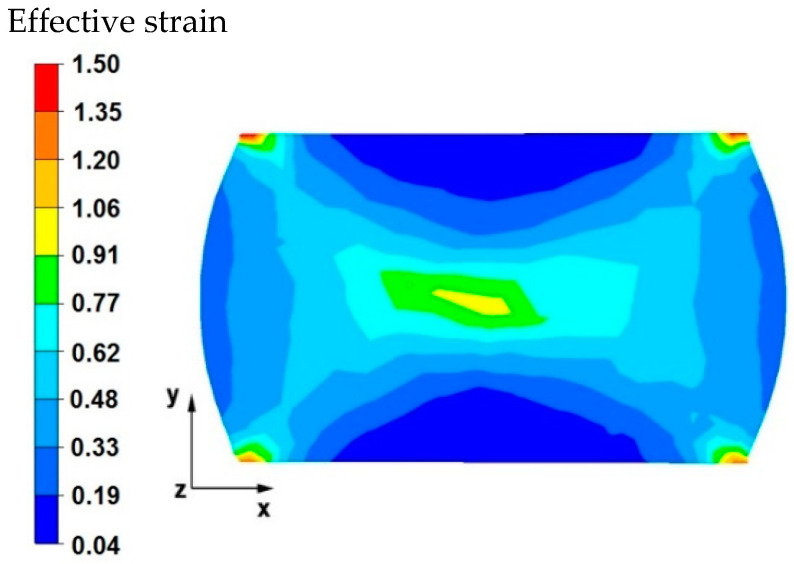
Distribution of the effective strain values on the surface of the cross-section of the sample with modeled discontinuity after deformation with 35% reduction.

**Figure 15 materials-15-07465-f015:**
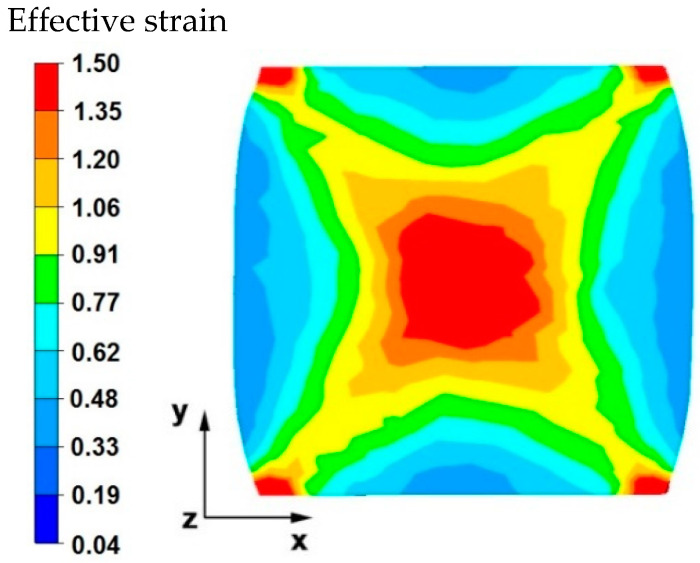
Distribution of the effective strain values on the surface of the cross-section of the sample with modeled discontinuity after rotation by 90° and re-deformation with 35% reduction.

**Figure 16 materials-15-07465-f016:**
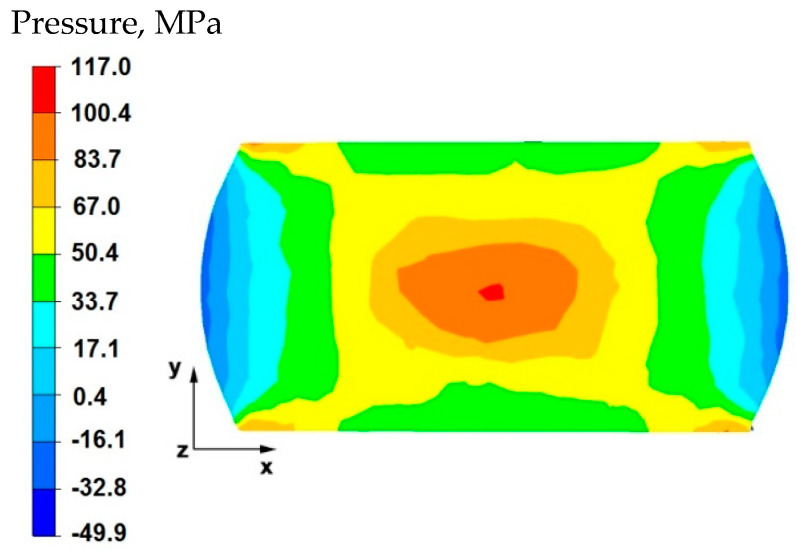
Distribution of hydrostatic pressure on the surface of the cross-section of the sample with modeled discontinuity after deformation with 35% reduction.

**Figure 17 materials-15-07465-f017:**
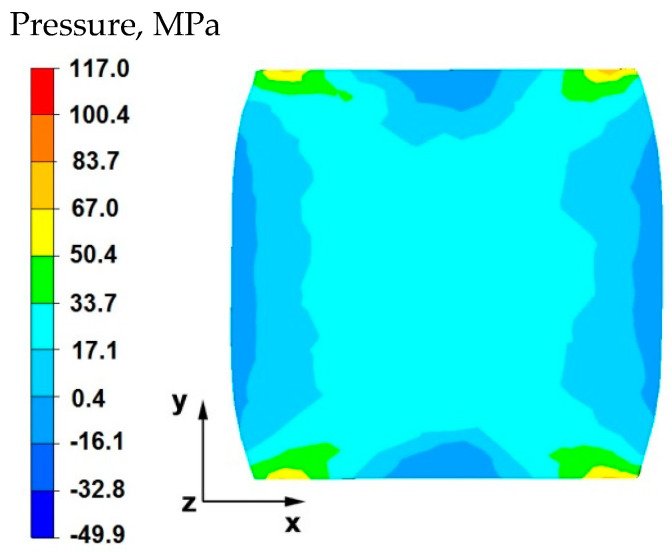
Distribution of hydrostatic pressure on the surface of the cross-section of the sample with modeled discontinuity after rotation by 90° and re-deformation with 35% reduction.

**Figure 18 materials-15-07465-f018:**
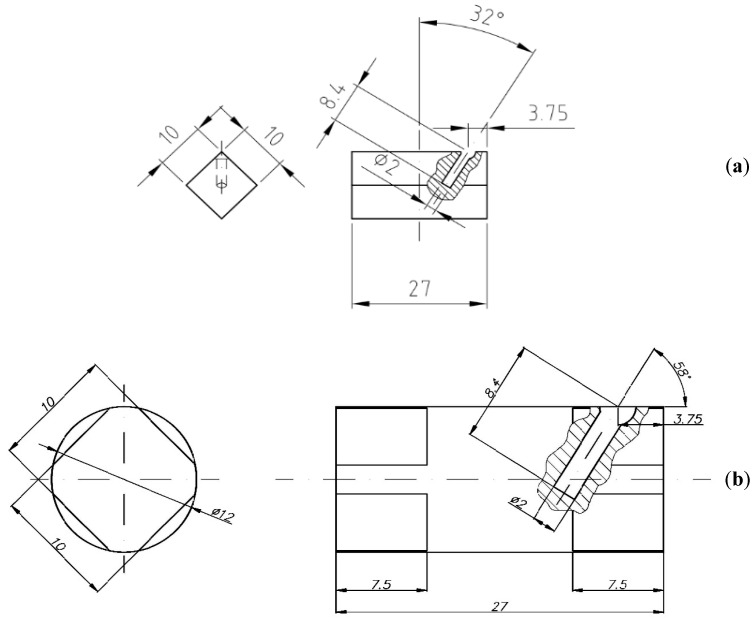
Samples prepared for the deformation process in the MaxStrain device: (**a**)—a sample with a square cross-section of 10 × 10 mm; (**b**)—a sample with a circular cross-section of 12 mm in diameter.

**Figure 19 materials-15-07465-f019:**
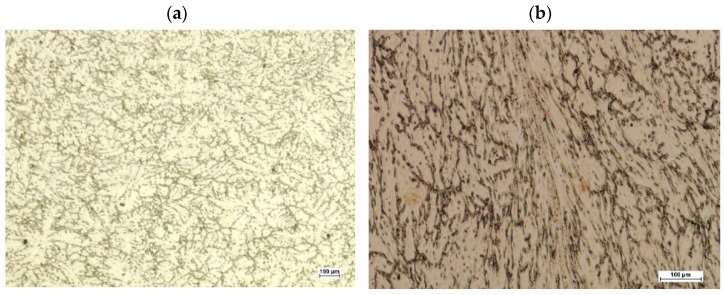
The disclosed microstructure of the sample with a square cross-section in the area of the axis after plastic deformation: (**a**)—sample without axial discontinuity; (**b**)—sample with artificially introduced axial discontinuity.

**Figure 20 materials-15-07465-f020:**
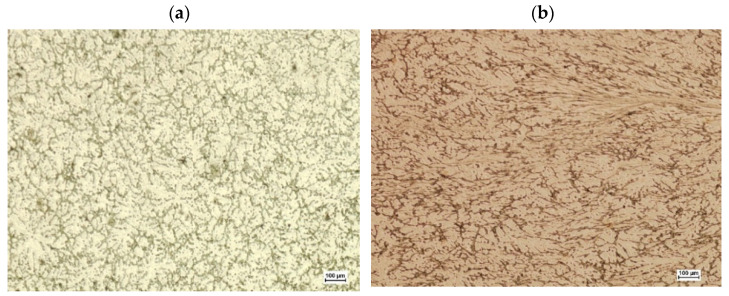
The disclosed microstructure of the sample with a circular cross-section in the area of the axis after plastic deformation: (**a**)—sample without axial discontinuity; (**b**)—sample with artificially introduced axial discontinuity.

**Figure 21 materials-15-07465-f021:**
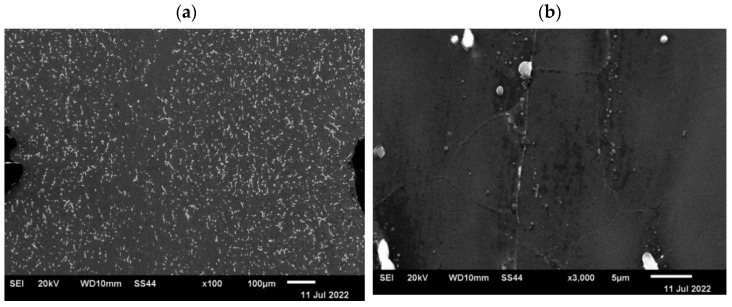
The microstructure of the sample with a square cross-section in the axis area after plastic deformation disclosed in the SEM analysis: (**a**)—magnification ×100; (**b**)—magnification ×3500.

**Figure 22 materials-15-07465-f022:**
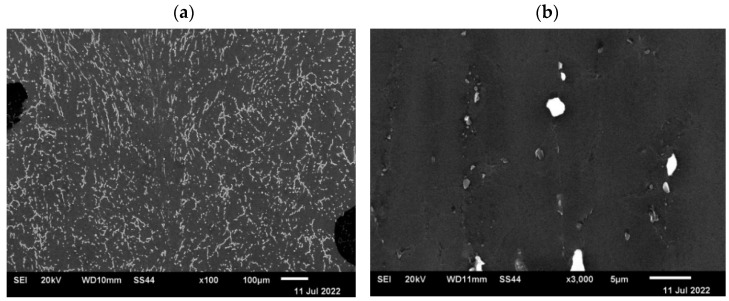
The microstructure of the sample with a circular cross-section in the area of the axis after plastic deformation disclosed in the SEM analysis: (**a**)—magnification ×100; (**b**)—magnification ×3500.

**Table 1 materials-15-07465-t001:** Chemical composition of the investigated alloy [%].

Alloy	Zn	Al	Si	Cu	Mn	Fe	Ni	Mg
AZ91	0.59	8.98	0.05	0.006	0.23	0.013	0.003	R

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
