# Peer review of "Modeling of the Closure of Metallurgical Defects in the Magnesium Alloy Die Forging Process"

_materials, 2022, doi:10.3390/ma15217465_

Round 1
Reviewer 1 Report
1.The aim of the study is presented clearly and in detail.
2. The presented results are new and important for application of materials. So the results can contribute to improving the properties of AZ91.and also AZ91 composites.
3. "The thermal calculation.." p.3, l.98 is incorrect. Calculation of thermal properties?
4. Figures should be better described.
5. Interpretations of results in Figures should be better described.
Author Response
Dear Editor and Reviewers,
We appreciate your valuable comments and detailed suggestions on our manuscript “Modelling of the closure of metallurgical defects in the magnesium alloy die forging process”, which was submitted to MATERIALS Special Issue
The revisions have been clearly highlighted using the "Track Changes" and “Comments” functions in Microsoft Word, therefore they are easily visible to editors and reviewers. The detailed explanations of all comments of the reviewers are provided underneath. We hope the revised manuscript can meet the requirements and high scientific standard of MATERIALS.
The aim of the study is presented clearly and in detail.
Response: The authors thank you for your comment
The presented results are new and important for application of materials. So the results can contribute to improving the properties of AZ91.and also AZ91 composites.
Response: The authors thank you for your comment
"The thermal calculation.." p.3, l.98 is incorrect. Calculation of thermal properties?
Response: Appropriate correction has been made in the text.
Figures should be better described.
Response: Appropriate correction has been made in the text.
Interpretations of results in Figures should be better described.
Response: Appropriate correction has been made in the text.
I would like to inform you that the manuscript was translated in a translation agency with which our university cooperates. As suggested by the reviewers, the final text was corrected by a native speaker.
Reviewer 2 Report
1. There are many figures in the paper, but they are not well organized and laid out, so it is recommended to merge the related images together. In addition, the range of the scale is as consistent as possible, so that it can be easily compared, such as Figure 1 and Figure 2, etc. Some figures are still missing the scale and need to be given.
2. This study focuses on the technology of hot working forming process of AZ91 alloy, nevertheless, it is suggested to further highlight the innovation in the introduction.
3. Although the authors have simulated the thermal processing and given typical microstructure pictures, they are not sufficient in terms of mechanism and discussion. It is also recommended to supplement EBSD and microhardness tests, which can further establish the intrinsic connection between process, defects, microstructure and properties.
Author Response
Dear Editor and Reviewers,
We appreciate your valuable comments and detailed suggestions on our manuscript “Modelling of the closure of metallurgical defects in the magnesium alloy die forging process”, which was submitted to MATERIALS Special Issue
The revisions have been clearly highlighted using the "Track Changes" and “Comments” functions in Microsoft Word, therefore they are easily visible to editors and reviewers. The detailed explanations of all comments of the reviewers are provided underneath. We hope the revised manuscript can meet the requirements and high scientific standard of MATERIALS.
Review
There are many figures in the paper, but they are not well organized and laid out, so it is recommended to merge the related images together. In addition, the range of the scale is as consistent as possible, so that it can be easily compared, such as Figure 1 and Figure 2, etc. Some figures are still missing the scale and need to be given.
Response: Appropriate correction has been made in the text.
This study focuses on the technology of hot working forming process of AZ91 alloy, nevertheless, it is suggested to further highlight the innovation in the introduction.
Response: Appropriate correction has been made in the text.
Although the authors have simulated the thermal processing and given typical microstructure pictures, they are not sufficient in terms of mechanism and discussion. It is also recommended to supplement EBSD and microhardness tests, which can further establish the intrinsic connection between process, defects, microstructure and properties.
Response: The study using scanning microscopy was of a an overview to show that metallurgical discontinuities remained closed. The aim of the study was not to analyze the structure, but to prove that the adopted method of plastic processing will enable the closure of metallurgical discontinuities. The authors plan to conduct research in the scope indicated by the reviewer in the future.
I would like to inform you that the manuscript was translated in a translation agency with which our university cooperates. As suggested by the reviewers, the final text was corrected by a native speaker.
Round 2
Reviewer 1 Report
The manuscript has been revised It improve and can be published
Reviewer 2 Report
I re-evaluated the revised manuscript and the author's response to the reviewer's comments. The authors have addressed the concerns raised by and manuscript quality has been increased.